# Emotional overeating affected nine in ten female students during the COVID-19 university closure: A cross-sectional study in France

**Aymery Constant**[1,2], **Alexandra Fortier**[1], **Yann Serrand**[1], **Elise Bannier**[3,4], **Romain Moirand**[1,5], **Ronan Thibault**[1,6], **Nicolas Coquery**[1], **Ambre Godet**[1], **David Val-Laillet**[1]*

**1** INRAE, INSERM, CHU Rennes, Nutrition Metabolisms and Cancer, NuMeCan, Univ Rennes, Rennes, France, **2** EHESP, School of Public Health, Rennes, France, **3** Inria, CRNS, Inserm, IRISA UMR 6074, Empenn U1228, Univ Rennes, Rennes, France, **4** Radiology Department, CHU Rennes, Rennes, France, **5** Unité d'Addictologie, CHU Rennes, Rennes, France, **6** Unité de Nutrition, CHU Rennes, Rennes, France

⦿ These authors contributed equally to this work.
‡ AG and DVL also contributed equally to this work.
* david.val-laillet@inrae.fr

**Data Availability Statement:** The datasets generated and analyzed for this study are available

## Abstract

### Objectives

To estimate the proportion of female university students reporting overeating (EO) in response to emotions during the COVID-19 university closures, and to investigate social and psychological factors associated with this response to stress.

### Design

Online survey gathered sociodemographic data, alcohol/drugs use disorders, boredom proneness and impulsivity using validated questionnaires, and EO using the Emotional Overeating Questionnaire (EOQ) assessing eating in response to six emotions (anxiety, sadness, loneliness, anger, fatigue, happiness), whose structure remains to be determined.

### Participants

Sample of 302 female students from Rennes University, France.

### Main outcome measure

Frequencies of emotional overeating.

### Analysis

The frequency of emotional overeating was expressed for each emotion as percentages. Exploratory Factor analyses (EFA) were used to determine EOQ structure and provide an index of all EOQ items used for further analysis. Linear regression models were used to explore relationships between EO and others covariates.

from the following address: https://doi.org/10.
57745/87KZFH.

**Funding:** The present research was funded by the
University of Rennes 1, Fondation de l'Avenir, the
Benjamin Delessert Institute, and INRAE. A. Godet
received a PhD grant from the University of Rennes
1. The funders had no role in study design, data
collection and analysis, decision to publish, or
preparation of the manuscript.

**Competing interests:** The authors have declared
that no competing interests exist.

## Results

Nine in ten participants reported intermittent EO in the last 28 days, mostly during 6 to 12
days, in response to Anxiety (75.5%), Sadness (64.5%), Happiness (59.9%), Loneliness
(57.9%), Tiredness (51.7%), and to a lesser extent to Anger (31.1%). EFA evidenced a one-
factor latent variable reflecting "Distress-Induced Overeating" positively correlated with
internal boredom proneness, tobacco use, attentional impulsivity, inability to resist emo-
tional cues, and loss of control over food intake, and negatively with age and well-being. EO
was unrelated to body mass index or substance abuse.

## Conclusion and implications

Nine in ten female students reported emotional overeating during the COVID-19 university
closure. This response to stress was related to eating tendencies typical of young women,
but also to personality/behavioral patterns such as boredom and impulsivity proneness. Bet-
ter understanding of the mechanisms underlying EO in response to stress and lack of exter-
nal/social stimulation would improve preventive interventions.

## Introduction

Studying at a university is a period of socialization for young adults, and studies show that con-
tacts with others positively influence well-being [1]. But in the early stages of the COVID-19
crisis, universities and other educational establishments switched from classroom to online
teaching, precluding social/external stimulations and support. Apart from a short period
between September and October 2021, universities across France remained fully closed
between March 2020 and August 2021, and students were already vulnerable to mental health
problems [2, 3] when they had to face this unprecedented situation, which drastically changed
their social and professional lives [4, 5]. As a result, French students showed higher depressive
symptoms than non-students during the first national lockdown (March-May 2020), compara-
ble rates during the easing phase (August-October 2020), and again dramatic increases during
the second lockdown (October-December 2020).

When individuals have difficulty adapting to stressful situations, they may display coping
responses susceptible to jeopardize their health and well-being [6], such as addictive and other
maladaptive behaviours [7]. In France, The COVID-19 pandemic and lockdown resulted in
frequent and mostly unhealthy changes in lifestyle among the general population, but addictive
behaviour such as drinking and smoking remained under control [8, 9]. When it comes to eat-
ing habits, however, findings from a systematic review of 23 studies indicated a shift towards
modified eating behaviours, characterized by an increased snack frequency and a preference
for sweets and ultra-processed food rather than fruits, vegetables, and fresh food [10]. Changes
in eating habits amid stressful situations may be related to "emotional eating", traditionally
defined as (over)eating in response to negative emotions [11]. This behaviour may occur as a
time-limited response to emotions in individuals without clinical condition [12], but may also
correlate with binge frequency, eating disorder features, and depression. From a clinical per-
spective, emotional eating has received increasing attention, particularly because a negative
emotional state is the most widely reported antecedent to binge eating episodes [11].

Young women could have been particularly vulnerable to emotional overeating during the
COVID-19 crisis. Firstly, they were more likely to report the negative impacts of COVID-19
on stress levels to be very much or an extreme amount compared to males, whereas males

were more likely to report the negative impacts to be not at all or a little compared to females [13]. Secondly, research suggests a gender difference in stress-related eating, with women choosing more palatable food [14, 15] and being more prone than men to turn to food for psychological comfort rather than physiological need [16–19]. Thirdly, one review highlighted that women have genetic predisposition for higher impulsivity and higher reward sensitivity, which are associated with dopamine dysregulation during comfort eating [20]. Impulsivity could be defined as a predisposition toward rapid, unplanned reactions to internal or external stimuli without regard to the negative consequences of these reactions [21], which has been found associated with stress eating [22, 23]. Finally, young women are more likely than men to eat while bored, especially when boredom-prone [24], and a review suggested that quarantine, reduced social and physical contacts with others were frequently shown to cause boredom [25]. Altogether, as women were more likely to report social isolation as being difficult or very difficult compared to men during the COVID-19 crisis [13], the COVID-19 university closures may have increased emotional eating in female students, particularly among those prone to boredom and/or impulsivity.

Characterizing emotional overeating during the COVID-19 pandemic university closure is of importance, since this coping response could further lead to addictive and health problems [26]. We chose female students as the sample of interest since they share personality/behavioural patterns that could increase the likelihood of emotional overeating during the COVID-19 university closures. The specific objectives of our study were to estimate the proportion of female university students reporting overeating (EO) in response to emotions during the COVID-19 university closures, and to investigate social and psychological factors associated with this response to stress.

## Materials and methods

### Participants and procedure

To meet the study objectives, we conducted an online survey between February and May 2021. Female students from the University of Rennes (France) aged 18–24 years who responded to a call for volunteers *via* students' mailing lists were eligible. A study number was attributed to each participant for pseudonymisation in a secured database to ensure confidentiality, as email addresses and personal information were recorded in a separate file in a locked computer. Only two researchers implicated in the study had access to the secured file, which was not accessible for the researcher who performed data analyses. In order to obtain consent, participants agreed to participate in the study by sending us an e-mail of acceptance, and then they received their pseudonymisation number and the link to complete the questionnaire. This reply email was then deleted. After free and informed written consent, participants completed questionnaires assessing psychological and behavioural variables, together with sociodemographic characteristics such as age (in year), weight (in kg), height (in cm), tobacco use (never, occasional, regular), and housing (living alone, with a partner, family or with roommates). Being aged less than 18 and more than 24 years old were the only non-inclusion criteria. Any medical information was excluded from data collection. The INRAE data protection agent approved the declaration of conformity of this online questionnaire study, which was used for the selection of volunteers to be included in a laboratory neurocognitive study conducted in the Rennes University Hospital and approved by an independent national research ethics committee under the supervision of the French Ministry of Higher Education, Research and Innovation (Comité de Protection des Personnes "Ile de France XI", project N˚21.02859.000020– 21071; N˚ID-RCB/EUDRACT 2021-A02314-37; National Clinical Trial number:

NCT05200182). This study was conducted in accordance of the principles of the Declaration of Helsinki.

## Measures

**Frequency of overeating episodes in response to emotions.** The Emotional Overeating Questionnaire (EOQ) is a six-item self-report questionnaire that assesses frequency of overeating episodes in response to six emotions, namely anxiety, sadness, loneliness, tiredness, anger, and happiness, previously used in a French study [27]. Its factor structure in French young women remains however to be determined using Exploratory Factor analyses (EFA). Each item begins with, "Have you eaten an unusually large amount of food given the circumstances in response to feelings of (. . .)". Each of the six emotions is presented in all capital letters, followed by three more synonyms in parentheses and in lower case. The response set for the six items is a 7-point scale reflecting the frequency of days on which the behaviour occurred in the past 28 days (*i.e.*, 0 = no days, 1 = 1–5 days, 2 = 6–12 days, 3 = 13–15 days, 4 = 16–22 days, 5 = 23–27 days, and 6 = every day).

**Alcohol or substance abuse.** The CRAFFT Screening Test consists of a series of six questions developed to screen adolescents for high-risk alcohol and other drug use disorders [28, 29]. It is a short, effective screening tool meant to assess whether a longer conversation about the context of use, frequency, and other risks and consequences of alcohol and other drug use is warranted. The questions are the following. 1) Have you ever ridden in a car driven by someone (including yourself) who was "high" or had been using alcohol or drugs? 2) Do you ever use alcohol or drugs to relax, feel better about yourself, or fit in? 3) Do you ever use alcohol/drugs while you are by yourself, alone? 4) Do you ever forget things you did while using alcohol or drugs? 5) Do your family or friends ever tell you that you should cut down on your drinking or drug use? 6) Have you gotten into trouble while you were using alcohol or drugs? A score of two positive answers or above indicates a potential drug issue. Diagnostic threshold for regular substance use in the French version of the CRAFFT was fixed at two positive answers with a sensitivity of 90.3% and a specificity of 77.7%.

**Boredom proneness.** The Boredom Proneness Scale (BPS) was developed by Farmer and Sundberg as a measure of the tendency to become bored. In the French validation study, factorial analyses yielded a two-factor structure including 26 items out of 28. The boredom proneness (BP) due to a lack of internal stimulation, or "internal BP" subscale includes 14 items related to one's inability to generate interesting activities, while the boredom proneness due to a lack of external stimulation, or "external BP" subscale includes 12 items related to the perception of low environmental/social stimulation. Internal-consistency reliability coefficients (Cronbach's α) for each of the two scales were above the 0.70 standard (α = 0.78 and 0.77, respectively). These two dimensions found support in the majority of studies on BP.

**Impulsivity.** The Barratt Impulsiveness Scale (BIS) is a 30-item self-report scale that is commonly used to measure impulsiveness and was validated in French [30]. Participants responded to each item using a 4-point frequency scale: 1 (rarely/never), 2 (occasionally), 3 (often), and 4 (almost always/always). Factor analysis revealed six primary factors of the scale: 1) attention (*e.g.*, "I am restless at the theatre or lectures"), 2) motor impulsiveness (*e.g.*, "I do things without thinking"), 3) self-control (*e.g.*, "I say things without thinking"), 4), cognitive complexity (*e.g.*, "I get easily bored when solving thought problems"), 5) perseverance (*e.g.*, "I change jobs"), and 6) cognitive instability (*e.g.*, "I have 'racing' thoughts"). Three secondary factors have been identified: attentional impulsiveness (mixture of primary factors 1 and 6), motor impulsiveness (mixture of primary factors 2 and 5), and non-planning impulsiveness (mixture of primary factors 3 and 4). Internal-consistency reliability coefficients (Cronbach's

α) for each of the three secondary factors were 0.70, 0.69 and 0.67, respectively. Attentional impulsiveness is defined as the difficulty focusing on a task at hand, motor (or behavioural) impulsiveness is defined as acting without thinking or on the spur of the moment, while non-planning impulsivity is characterized as present-moment focus without regard for future consequences.

**Well-being.** The World Health Organization Well-Being Index (WHO-5) is a 5-item questionnaire assessing subjective psychological well-being in research and clinical settings, and available in 30 languages including French. The WHO-5 consists of five statements, which respondents rate according to a frequency scale (0 = at no time; 5 = all the time) indicative of positive mood (good spirits, relaxation), vitality (being active and waking up fresh and rested), and general interest (being interested in things). Item response theory analyses in studies of younger persons and elderly persons indicate that the measure has good construct validity as a unidimensional scale measuring well-being in these populations [31]. The total raw score, ranging from 0 to 25, is multiplied by 4 to give the final score, with 0 representing the worst imaginable well-being and 100 representing the best imaginable well-being. A score below 50 can indicate poor well-being which may be secondary to a depressive disorder or other etiology and is an indication for further evaluation.

**Cognitive and behavioural components of eating.** According to another study [32], we used the Three-Factor Eating Questionnaire Revised, 18-item (TFEQ-R18) to measure cognitive and behavioural components of eating. Three subscales are included in this questionnaire: (1) Cognitive Restraint (conscious restriction of food intake in order to control body weight or to promote weight loss) comprised of six items (e.g., "I consciously hold back at meals in order not to gain weight), (2) Uncontrolled Eating (tendency to eat more than usual due to a loss of control over intake accompanied by subjective feelings of hunger), comprised of nine items (*e.g.*, "When I see a real delicacy, I often get so hungry that I have to eat right away"), and (3) Emotional Eating (inability to resist emotional cues), comprised of three items (*e.g.*, "When I feel blue, I often overeat"). Internal-consistency reliability coefficients (Cronbach's α) for each of the three scales were above the 0.70 standard and below the 0.90 limit recommended for individual assessment. Responses are scored on a 4-point scale and anchors can vary across items (*e.g.*, definitely true to definitely false, or never to at least once a week). The raw scale scores are transformed to a 0–100 scale. Higher scores in the respective scales are indicative of greater cognitive restraint, uncontrolled, or emotional eating.

## Statistical analyses

We used the same statistical approach as in our previous work on the same topic/population [12]. Categorical data were expressed as numbers (N) and percentages (%) and numerical data were expressed as means (M) and standard deviations (±SD). An exploratory factor analysis was performed on the EOQ items using an Unweighted Least-Square factor analysis. This method was found to provide accurate and conservative parameter estimates when using ordinal data. A Promax rotation, a non-orthogonal (oblique) solution in which the factors are allowed to be correlated followed the factor. in order to establish which of the 6 items in EOQ belonged to latent domains or conceptual areas and which items should be maintained in factor scores. Factor scores are composite linear variables which provide information about an individual's placement on the factor(s), and can be used as an index of EOQ items for further analysis. A generalized linear regression model to estimate the strength of the association between emotional overeating and each covariate (univariate analysis) as the factor score was a linear variable. In order to determine variables independently associated with EO, all variables that demonstrated an association with p < 0.05 in the univariate analysis were considered as

candidate variables in the multiple linear regression model. The pseudo R-squared is not discussed in generalized linear models texts [33]. Statistical analyses were performed using the SPSS statistical package, version 19 (SPSS, Chicago, Illinois, United States).

## Results

### Descriptive analysis

Survey questionnaires were filled by 320 university women students between February and May 2021, but 18 (4.8%) were rejected because of missing data in EOQ. The 302 remaining respondents were aged 20.9 years on average (Table 1), with a minority living alone (37.4%). The mean Body Mass Index (±SD) was 21.7±3.4, and most of participants were normal-weight students (75.8%) while 11.3% were underweight and 12.9% had overweight/obesity. Standardized subscales mean scores (ranging from 0 to 100) of the TFEQ-R18 were 37.0±24.4 for Cognitive Restraint, 43.1±22.4 for Uncontrolled Eating, and 55.5±30.9 for Emotional Eating. Boredom Proneness (BP) Standardized subscales mean scores were 29.2±20.8 for internal and

**Table 1. Characteristics of the respondents (N = 302).**

| Variables | N (%) |
|---|---|
| Age | 20.9±1.8 |
| Living condition | |
| Flatshare | 61 (20.2) |
| In family | 74 (24.5) |
| In couple | 54 (17.9) |
| Alone | 113 (37.4) |
| BMI (mean±SD) index | 21.7±3.4 |
| Weight status (BMI) | |
| Obesity (>30) | 8 (2.6) |
| Overweight (25–29.9) | 31 (10.3) |
| Normal (18–24.9) | 229 (75.8) |
| Underweight (<18) | 34 (11.3) |
| Tobacco use | 40 (13.2) |
| Boredom proneness (range 0–100) † | |
| Internal | 29.2±20.8 |
| External | 44.9±21.2 |
| Impulsivity (range 0–100) † | |
| Attentional | 39.9±15.5 |
| Motor | 28.4±10.1 |
| Non planning | 37.9±13.1 |
| WHO Well-being score (range 0–100) | 55.8±19.8 |
| Alcohol/Substance abuse (CRAFTT) | 44 (14.6) |
| Components of Eating (TFEQ-R18)† | |
| Cognitive restraint | 37.0±24.4 |
| Uncontrolled eating | 43.1±22.4 |
| Emotional eating | 55.5±30.9 |

Note: BMI = Body Mass Index; SD = standard deviation.

†: The raw scale scores were transformed to a 0–100 scale [((raw score − lowest possible raw score)/possible raw score range) × 100]. Higher scores in the respective scales are indicative of greater cognitive restraint, uncontrolled, or emotional eating.

44.9±21 for external BP, while Barratt Impulsiveness Scale (BIS) Standardized subscales mean scores were 39.9±15.5 for attentional, 28.4±10.1 for motor, and 37.9±13.12 for non-planning impulsivity.

## Emotional overeating frequency in the last 28 days

Nine in ten respondents (91.4%) reported Emotional overeating episodes during at least 1–5 days in the past month. EO episodes reached at least 6 days in 60.3% of cases, and 13–15 days in 38.4% of cases. One in four (24.8%) reported EO episodes exceeding 15 days in the last 28-day period. They reported emotional overeating episodes mostly in response to anxiety (75.5%), sadness (64.9%), happiness (59.9%), loneliness (57.9%) and tiredness (51.7%). Only 31.1% of them reported OE episodes in response to anger (Table 2). Unweighted Least-Square exploratory factor analysis followed by a Promax rotation was performed on the 6 EOQ items. After the first rotation, the "Happiness" item was removed because it loaded relatively low (< 0.40) on the two-factor solution (Eigen values > 1). A final extraction was performed on the remaining 5 items (Anxiety; Loneliness; Sadness; Anger; and Fatigue), resulting in a one-factor solution explaining 66% of the common variance of the data (Table 3). This latent variable was interpreted as "Distress-Induced Overeating" (DIO), and a factor score was computed and used as dependent variable in the multivariate analysis. Frequency of overeating episodes in response to Happiness was used as a single outcome.

## Multivariate analysis

In univariate analysis (Table 4; Univariate estimates), the DIO factor score was positively associated with tobacco use, boredom proneness, substance abuse, inability to resist emotional cues (EE), and loss of control over food intake (uncontrolled eating, UE). It was negatively associated with age and well-being. In multivariate analysis (Table 4; Multivariate estimates), the DIO factor score was positively associated with tobacco use, boredom proneness, the inability to resist emotional cues (EE), and loss of control over food intake (uncontrolled eating, UE). It was negatively associated with age. In univariate and multivariate analyses, overeating in response to happiness (Table 5) was positively associated with well-being and uncontrolled eating, and more frequent in participant living in family as compared to those living alone.

## Discussion

Our results showed that 9 in 10 female students included in our study reported intermittent Emotional Overeating in the last 28 days, mostly during 6 to 12 days, in response to Anxiety (75.5%), Sadness (64.5%), Happiness (59.9%), Loneliness (57.9%), Tiredness (51.7%), and to a lesser extent in response to Anger (31.1%). Exploratory factor analysis evidenced a one-factor

**Table 2. Emotional eating days in the last 28 days in response to six emotional states among the study sample (N = 302).** Data are expressed as Number and Percentages (%).

| | 0 day | 1–5 days | 6–12 days | 13–15 days | >16 days |
|---|---|---|---|---|---|
| Anxiety | 74 (24.5) | 115 (38.1) | 44 (14.6) | 19 (6.3) | 50 (16.6) |
| Sadness | 106 (35.1) | 101 (33.4) | 39 (12.9) | 23 (7.6) | 33 (10.9) |
| Loneliness | 127 (42.1) | 87 (28.8) | 39 (12.9) | 23 (7.6) | 26 (8.6) |
| Anger | 208 (68.9) | 65 (21.5) | 13 (4.3) | 4 (1.3) | 13 (4.0) |
| Tiredness | 146 (48.3) | 80 (26.5) | 25 (8.3) | 15 (5.0) | 36 (11.9) |
| Happiness | 121 (40.1) | 87 (28.8) | 37 (12.3) | 27 (8.9) | 30 (9.9) |

**Table 3. EOQ items and factor loadings for the two- and the one-factor solutions.**

| EOQ Items | First rotation † | | 1 factor solution ‡ |
|---|---|---|---|
| | **Factor 1** | **Factor 2** | |
| *Anxiety* | 0.72 | 0.11 | 0.70 |
| *Sadness* | 0.89 | -0.28 | 0.91 |
| *Loneliness* | 0.79 | -0.35 | 0.79 |
| *Anger* | 0.67 | -0.07 | 0.69 |
| *Tiredness* | 0.76 | 0.34 | 0.69 |
| *Happiness* | 0.44 | 0.51 | |
| Eigen value | 3.50 | 1.01 | 3.3 |
| % of variance | 58.3 | 16.7 | 66.0 |
| Cronbach Alpha | | | 0.86 |

†: Rotated factor loadings following unweighted least square extraction and oblique (promax) rotation; ‡ Unrotated factor loadings following unweighted least square extraction, after the "Happiness" item was removed.

**Table 4. Factors associated with distress-induced overeating factorial score, generalized linear model.**

| Variables | Univariate estimates | | Multivariate model | |
|---|---|---|---|---|
| | **B** | **p-value** | **B** | **p-value** |
| Age | -0.085 | 0.006 | -0.072 | 0.004 |
| Living condition | | | | |
| Flat share | -0.112 | 0.458 | | |
| In family | -0.007 | 0.963 | | |
| In couple | -0.043 | 0.785 | | |
| Alone | Ref | | | |
| Body Mass Index | | | | |
| >25 | 0.203 | 0.215 | | |
| <18 | -0.114 | 0.511 | | |
| 18–24.9 | Ref | | | |
| Boredom proneness | | | | |
| Internal | 0.146 | <0.001 | 0.070 | <0.001 |
| External | 0.147 | <0.001 | 0.027 | 0.250 |
| Impulsiveness | | | | |
| Attentional | 0.075 | <0.001 | 0.029 | 0.038 |
| Motor | 0.017 | 0.410 | | |
| Non Planning | 0.046 | <0.001 | -0.001 | 0.903 |
| Tobacco use | | | | |
| Yes | 0.444 | 0.006 | 0.325 | 0.014 |
| No | Ref | | Ref | |
| WHO Well-being score | -0.018 | <0.001 | -0.007 | 0.130 |
| Alcohol/Substance abuse (CRAFTT) | | | | |
| Yes | 0.353 | 0.024 | 0.003 | 0.772 |
| No | ref | | Ref | |
| Components of Eating (TFEQ-R18) | | | | |
| Cognitive restraint | -0.023 | 0.761 | | |
| Uncontrolled eating | 0.064 | <0.001 | 0.033 | 0.001 |
| Emotional eating | 0.141 | <0.001 | 0.046 | 0.029 |

**Table 5. Factors associated with the overeating frequency in response to happiness, generalized linear model.**

| Variables | Model 1 | | Model 2 | |
|---|---|---|---|---|
| | Estimate | p-value | Estimate | p-value |
| Age | -0.078 | NS | | |
| Living condition | | | | |
| Flat share | -0.196 | NS | -0.155 | NS |
| In family | 0.578 | 0.009 | 0.547 | 0.011 |
| In couple | 0.421 | NS | 0.418 | 0.079 |
| Alone | Ref | | Ref | |
| Weight status | | | | |
| Overweight /obesity | 0.058 | NS | | |
| Underweight | -0.301 | NS | | |
| Normal weight | Ref | | | |
| Boredom proneness | | | | |
| Internal | -0.007 | NS | | |
| External | -0.035 | NS | | |
| Tobacco use | | | | |
| Yes | 0.404 | NS | | |
| No | Ref | | | |
| WHO Well-being score | 0.013 | 0.002 | 0.014 | 0.001 |
| Impulsivity | | | | |
| Attentional | 0.028 | NS | | |
| Motor | 0.062 | 0.017 | | |
| Planning | 0.061 | 0.002 | | |
| Alcohol/Substance abuse (CRAFTT) | | | | |
| Yes | -0.242 | NS | | |
| No | ref | | | |
| Components of Eating (TFEQ-R18)† | | | | |
| Cognitive restraint | 0.016 | NS | | |
| Uncontrolled eating | 0.047 | 0.001 | 0.051 | <0.001 |
| Emotional eating | 0.020 | NS | | |

latent variable reflecting "Distress-Induced Overeating" (DIO) including all EOQ items except Happiness, as previously described in a comparable population before the COVID-19 pandemic [12]. In multivariate analysis, the DIO factor score correlated positively with internal boredom proneness, tobacco use, attentional impulsivity, inability to resist emotional cues, and loss of control over food intake. It correlated negatively with age and well-being. Overeating in response to happiness correlated positively with living in family, well-being and loss of control over food intake. Overeating was not significantly related to BMI or substance abuse.

Studies on the effects of the pandemic on student mental health showed significant levels of stress, anxiety, depressive symptoms, concerns for oneself and one's family's health, reduced social interactions, and increased concerns over academic achievements [34]. Accordingly, the standardized well-being score in our study population was barely above the average of 50 (over 100) in our participants. In normal situations, students may use social and physical activities to cope with stress [35], but the reduction in social (collective training sessions or sport events) and physical (restricted access to exercise facilities, sport grounds and parks) opportunities to exercise increased sedentary behaviour [8]. The proportion of female student reporting alcohol

or substance abuse (12.2%) or regular smoking (7.2%) was however quite low in our study sample and similar to pre-pandemic levels [12]. While addictive disorders remained relatively under control, it seems that the COVID-19 crisis affected deeply eating behaviours in female students, since 91.4% of female students reported emotional overeating in the last month against half of the female students from the same university surveyed three years before [12]. Environmental constraints are likely to influence the type of coping strategies available such as social drinking, given that access to pubs and outdoor gatherings were restricted. And overeating was perhaps perceived as a coping response safer than alcohol consumption in female students continuing their courses and programs online amid social disruptions, which would need to be confirmed by specific questionnaires on individual motivation and perception. These behaviours were however not significantly related to short-term weight outcomes since the proportion of female students with overweight/obesity in our study sample was similar to national estimates prior the COVID-19 crisis [36].

Previous studies have underscored the specific role of anxiety on overeating [37–39], but descriptive and factor analyses indicated that sadness and loneliness also contributed greatly to distress-induced overeating (DIO) in our participants, reflecting the particular situations of students working remotely. In addition, personality/behavioural patterns typical of our French young women such as uncontrolled eating (UE) and inability to resist emotional cues (EE) were also related to DIO, together with BP and impulsivity. In univariate analysis, DIO had the strongest independent association with internal and external BP. Under non-pandemic circumstances, boredom-prone individuals tend to experience varying degrees of hopelessness, loneliness, distractibility, lack of motivation, and general dissatisfaction, and may use unhealthy and potentially addictive behaviours as coping mechanisms [24, 40–42]. It must be noted that eating in response to happiness was related but distinct from DIO, and seemed to influence food intake in relation to normal (*i.e.*, uncontrolled eating, well-being, and family life) rather than pathological forms of eating tendencies [43, 44].

In multivariate analysis, our findings suggest that emotional overeating was related to the inability to generate interesting activities during the pandemic in a context of limited social/external stimulations. This is in line with a study conducted in France showing that failure in engaging in a creative activity to overcome uncertainty and solitude fostered responses not requiring special or creative skills, such as overeating, particularly in women [45]. When it comes to impulsivity, higher impulsivity scores in healthy normal-weight women tend to predict higher food intake [46], and attentional impulsivity (*i.e.*, the difficulty focusing on a task at hand) is consistently related to various measures related to overeating, because of attention diverted to palatable food [47]. In contrast, non-planning impulsivity, when the immediately available small reward is preferentially chosen over a delayed larger reward [48], seems to be only weakly related to overeating [49]. Our results are in line with these studies, as only attentional impulsivity was related to DIO in the multivariate analysis. Finally, DIO seemed more frequent in smokers, which may correlate with indicators of academic stress [50], and less frequent in older students.

This study must be interpreted in light of its limitations. First, the cross-sectional design did not allow determining causal inferences about relationships between Emotional Overeating and other covariates under investigation, although personality/behavioural patterns theoretically precede current behaviours. Better understanding of the interactions between stress, coping, personality/behavioural patterns, emotional overeating and the risk of diseases, such as metabolic diseases, overweight and obesity, among students warrants a prospective study and follow-up assessments over the university year or cycle. Second, the EOQ has a single item for assessing each emotion-related eating. Although the factor structure and psychometrics properties of the EOQ were investigated in the present study, it still warrants a full validation

study including diverse subgroups from the French general population. Finally, eating and psychological disorders, which may be influential on emotional overeating, could Third, some behavioural factors related to emotional overeating (*e.g.* sleep and physical activity) were not assessed, and BMI estimates could have been biased due to self-reported data. Finally, we must reckon that the 302 respondents included in this study represented only a small percentage of the whole student population in our target University, but our recruitment campaign was performed in only one campus of the University and only female volunteers aged 18-24yo who expressed an interest in this study were recruited. To note, comparable descriptive studies were previously published with a similar number of volunteers [12].

## Implications for research and practice

Nine in ten female students reported emotional overeating during the COVID-19 university closure. This response to stress was related to eating tendencies typical of young women, such as uncontrolled eating or inability to resist emotional cues, but also to personality/behavioural patterns such as boredom and impulsivity proneness. In terms of perspective, a better understanding of the attentional, neurobiological and neurocognitive mechanisms underlying emotional eating in response to stress/emotions and lack of external/social stimulation would improve preventive interventions related to disordered eating in women coping with stress and/or isolation [51, 52].

## Author Contributions

**Conceptualization:** Nicolas Coquery, Ambre Godet, David Val-Laillet.

**Data curation:** Aymery Constant, Alexandra Fortier, Yann Serrand, Nicolas Coquery, Ambre Godet.

**Formal analysis:** Aymery Constant, Alexandra Fortier, Yann Serrand, Nicolas Coquery, Ambre Godet.

**Funding acquisition:** Nicolas Coquery, David Val-Laillet.

**Investigation:** Alexandra Fortier, Nicolas Coquery, Ambre Godet, David Val-Laillet.

**Methodology:** Aymery Constant, Nicolas Coquery, Ambre Godet, David Val-Laillet.

**Project administration:** Nicolas Coquery, David Val-Laillet.

**Resources:** David Val-Laillet.

**Supervision:** Elise Bannier, Romain Moirand, Nicolas Coquery, David Val-Laillet.

**Validation:** Aymery Constant, Nicolas Coquery, David Val-Laillet.

**Visualization:** Yann Serrand, Romain Moirand, Ronan Thibault, Nicolas Coquery, Ambre Godet, David Val-Laillet.

**Writing – original draft:** Aymery Constant.

**Writing – review & editing:** Yann Serrand, Elise Bannier, Romain Moirand, Ronan Thibault, Nicolas Coquery, Ambre Godet, David Val-Laillet.

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
