## [Decision Letter · Decision Letter 0]

10 Jul 2023

PONE-D-23-13686Emotional overeating affected nine in ten female students during the COVID-19 University closure: A cross-sectional study in FrancePLOS ONE

Dear Dr. Val-Laillet,

Thank you for submitting your manuscript to PLOS ONE. After careful consideration, we feel that it has merit but does not fully meet PLOS ONE’s publication criteria as it currently stands. Therefore, we invite you to submit a revised version of the manuscript that addresses the points raised during the review process.Kindly address the reviewers' suggestions.

We look forward to receiving your revised manuscript.

Kind regards,

Roberto Ariel Abeldaño Zuñiga

Academic Editor

PLOS ONE

- https://hal.science/hal-01856971v1/document

- https://doi.org/10.3389/fnut.2022.920170

- https://doi.org/10.1186/s12889-021-10531-3

In your revision ensure you cite all your sources (including your own works), and quote or rephrase any duplicated text outside the methods section. Further consideration is dependent on these concerns being addressed.

3. "Thank you for stating the following financial disclosure:

“The present research was funded by the University of Rennes 1, Fondation de l’Avenir, the Benjamin Delessert Institute, and INRAE. A. Godet received a PhD grant from the University of Rennes 1.”

Please include this amended Role of Funder statement in your cover letter; we will change the online submission form on your behalf."

Additional Editor Comments:

Dear authors,

Kindly address the reviewers' suggestions

Reviewers' comments:

Reviewer's Responses to Questions

**Comments to the Author**

1. Is the manuscript technically sound, and do the data support the conclusions?

Reviewer #1: Yes

Reviewer #2: Yes

Reviewer #3: Yes

2. Has the statistical analysis been performed appropriately and rigorously? 

Reviewer #1: Yes

Reviewer #2: Yes

Reviewer #3: Yes

3. Have the authors made all data underlying the findings in their manuscript fully available?

Reviewer #1: Yes

Reviewer #2: No

Reviewer #3: Yes

4. Is the manuscript presented in an intelligible fashion and written in standard English?

Reviewer #1: Yes

Reviewer #2: Yes

Reviewer #3: Yes

5. Review Comments to the Author

Reviewer #1: INTRODUCTION: in line 74 where it is stated: “Thirdly, women have genetic predisposition for higher impulsivity and higher reward sensitivity, which are associated with dopamine dysregulation during comfort eating”.

I suggest substantiating the assertion, using more studies supporting it, or in any case affirm that this was observed in “one” particular study, instead of generalizing.

DISCUSSION: in line 308 it is affirmed: “Finally, DIO seemed more frequent in smokers and less frequent in younger students, which may correlate with indicators of academic stress”

I suggest reviewing this affirmation, because the linear regression analysis shows a negative association between DIO and age

Reviewer #2: I suggest the following aspects be completed or appear in the article:

It does not appear as a criterion for non-inclusion of participants, if they having a diagnosis of depression or antidepressant treatment or sleeping pills.

It isn't clear whether physical activity and hours of sleep were evaluated as a control variable, it is important because they could be different in each participant and may affect the study variables.

The results do not present the R2 of the regression models, it is important to include the data to know the fit of the model.

The limitations of the study are recognized, but the inaccuracy of the weight and height measurements to calculate the BMI must be recognized, this could be under or over estimated, also if the control variables of physical activity, diagnosis of depression or treatment were not considered with antidepressants or sleeping pills, you should clarify it in the limitations.

Reviewer #3: The objective of the present investigation was to estimate the proportion of college women who reported overeating in response to emotions during the COVID-19 closure of universities, and to investigate the social and psychological factors associated with this stress response. This topic is extremely important, firstly because there are few studies that deal with the psychological state and secondly because this population already tends to have various affectations at this level.

To obtain their data, they applied an online questionnaire, which is a good method to obtain data remotely in an isolation situation, such as the COVID-19 pandemic.

The authors use adequate and validated tools for the population studied.

The University of Rennes has an enrollment of approximately 30,000 students, the authors recruited 300 students, this is one of the limitations of the study.

This study is important as it confirms that isolation due to the COVID-19 pandemic had a negative impact at different levels, especially in the psychological aspect. That 9 out of 10 students reported emotional overeating behaviors is alarming. This study will help to design interventions aimed at improving the psychological state of students, since many of the disorders adopted during the isolation due to the pandemic are still maintained today

6. PLOS authors have the option to publish the peer review history of their article (what does this mean?). If published, this will include your full peer review and any attached files.

Reviewer #1: **Yes: **Graciela F Scruzzi

Reviewer #2: No

Reviewer #3: **Yes: **Sabina López Toledo

---

## [Author Response · Author response to Decision Letter 0]

13 Jul 2023

#1 Response to the editor

ANSWER: Heading levels were corrected according to the submission guidelines, as well as table’s title. Manuscript’s text was double-spaced. A short title (<100 characters) was added. The names of the files were modified to fit with the standards.

- https://hal.science/hal-01856971v1/document

- https://doi.org/10.3389/fnut.2022.920170

- https://doi.org/10.1186/s12889-021-10531-3

In your revision ensure you cite all your sources (including your own works), and quote or rephrase any duplicated text outside the methods section. Further consideration is dependent on these concerns being addressed.

ANSWER: These three references were added in the manuscript and rephrasing was made in some instances, except for the methods where the appropriate quotation was made without deeply modifying the text. The first two references were cited on the M&M section while the third one was added in the introduction, where evoking the higher prevalence of emotional eating in women compared to men.

3. "Thank you for stating the following financial disclosure:

“The present research was funded by the University of Rennes 1, Fondation de l’Avenir, the Benjamin Delessert Institute, and INRAE. A. Godet received a PhD grant from the University of Rennes 1.”

Please include this amended Role of Funder statement in your cover letter; we will change the online submission form on your behalf."

ANSWER: The following statement was added in the funding section and in the cover letter: “The funders had no role in study design, data collection and analysis, decision to publish, or preparation of the manuscript.”

ANSWER: The data are already available and we provided the corresponding weblink in the “Data Availability Statement” section (p.26): “The datasets generated and analyzed for this study are available at the following address: https://doi.org/10.57745/87KZFH”

ANSWER: Two studies (Brooks et al. 2020 and Topp et al., 2015) were corrected in the required citing style, and three other studies were added in the reference list: Coquery et al. (2022), McCullagh, P., & Nelder, J.A (1989) and Sze et al. (2021).

2. Review Comments to the Author

Reviewer #1: INTRODUCTION: in line 74 where it is stated: “Thirdly, women have genetic predisposition for higher impulsivity and higher reward sensitivity, which are associated with dopamine dysregulation during comfort eating”.

I suggest substantiating the assertion, using more studies supporting it, or in any case affirm that this was observed in “one” particular study, instead of generalizing.

ANSWER: This assumption was based on a review article; we accordingly added this precision, thank you for the clarification. 

DISCUSSION: in line 308 it is affirmed: “Finally, DIO seemed more frequent in smokers and less frequent in younger students, which may correlate with indicators of academic stress”

I suggest reviewing this affirmation, because the linear regression analysis shows a negative association between DIO and age

ANSWER: Thank you for highlighted this mistake, the sentence was corrected accordingly “Finally, DIO seemed more frequent in smokers, which may correlate with indicators of academic stress (Nichter et al. 2007), and less frequent in older students.” (p.19)

Reviewer #2: I suggest the following aspects be completed or appear in the article:

It does not appear as a criterion for non-inclusion of participants, if they having a diagnosis of depression or antidepressant treatment or sleeping pills.

ANSWER: Thank you for your comment. This precision was added in the manuscript lines [105-106] “Being aged less than 18 and more than 24 years old were the only non-inclusion criteria. Any medical information was excluded from data collection.” 

It isn't clear whether physical activity and hours of sleep were evaluated as a control variable, it is important because they could be different in each participant and may affect the study variables.

ANSWER: These variables were not controlled in the present study. We added this statement in the limits section lines [326]. “Third, some behavioural factors related to emotional overeating (e.g. sleep and physical activity) were not assessed”

The results do not present the R2 of the regression models, it is important to include the data to know the fit of the model.

ANSWER: The R2 are not indicated nor interpreted in the generalized linear models (GLM). A sentence and reference have been added accordingly in the M&M: “The pseudo R-squared is not discussed in generalized linear models texts (McCullagh & Nelder, 1989).”

The limitations of the study are recognized, but the inaccuracy of the weight and height measurements to calculate the BMI must be recognized, this could be under or over estimated, also if the control variables of physical activity, diagnosis of depression or treatment were not considered with antidepressants or sleeping pills, you should clarify it in the limitations.

ANSWER: We added this precision following the limitations of the lack of reported physical activity and sleeping information, line [327] “, and BMI estimates could have been biased due to self-reported data.”

Reviewer #3: The objective of the present investigation was to estimate the proportion of college women who reported overeating in response to emotions during the COVID-19 closure of universities, and to investigate the social and psychological factors associated with this stress response. This topic is extremely important, firstly because there are few studies that deal with the psychological state and secondly because this population already tends to have various affectations at this level.

To obtain their data, they applied an online questionnaire, which is a good method to obtain data remotely in an isolation situation, such as the COVID-19 pandemic.

The authors use adequate and validated tools for the population studied.

The University of Rennes has an enrollment of approximately 30,000 students, the authors recruited 300 students, this is one of the limitations of the study.

This study is important as it confirms that isolation due to the COVID-19 pandemic had a negative impact at different levels, especially in the psychological aspect. That 9 out of 10 students reported emotional overeating behaviors is alarming. This study will help to design interventions aimed at improving the psychological state of students, since many of the disorders adopted during the isolation due to the pandemic are still maintained today

ANSWER: Thank you for your comment. Even though we reckon that the potential of recruitment at the University of Rennes 1 is around 30,000 male and female students, it is important to precise that our recruitment campaign was performed in only one campus of the University (Beaulieu) and that only female volunteers aged 18-24yo who expressed an interest in this study were recruited. Previous descriptive studies have been performed and published with a similar number of volunteers (e.g. Constant et al. 2018).

---

## [Decision Letter · Decision Letter 1]

15 Aug 2023

Emotional overeating affected nine in ten female students during the COVID-19 University closure: A cross-sectional study in France

PONE-D-23-13686R1

Dear Dr. Val-Laillet,

We’re pleased to inform you that your manuscript has been judged scientifically suitable for publication and will be formally accepted for publication once it meets all outstanding technical requirements.

Kind regards,

Roberto Ariel Abeldaño Zuñiga

Academic Editor

PLOS ONE

Additional Editor Comments (optional):

Reviewers' comments:

Reviewer's Responses to Questions

**Comments to the Author**

1. If the authors have adequately addressed your comments raised in a previous round of review and you feel that this manuscript is now acceptable for publication, you may indicate that here to bypass the “Comments to the Author” section, enter your conflict of interest statement in the “Confidential to Editor” section, and submit your "Accept" recommendation.

Reviewer #1: All comments have been addressed

Reviewer #2: All comments have been addressed

2. Is the manuscript technically sound, and do the data support the conclusions?

Reviewer #1: Yes

Reviewer #2: Yes

3. Has the statistical analysis been performed appropriately and rigorously? 

Reviewer #1: Yes

Reviewer #2: Yes

4. Have the authors made all data underlying the findings in their manuscript fully available?

Reviewer #1: Yes

Reviewer #2: Yes

5. Is the manuscript presented in an intelligible fashion and written in standard English?

Reviewer #1: Yes

Reviewer #2: Yes

6. Review Comments to the Author

Reviewer #1: I believe that this article is ready to be published, it makes a contribution to the changes in diet during the pandemic. In addition, my suggestions were taken into account and carried out

Reviewer #2: I confirm that the observations I made have been clarified and incorporated in the corresponding sections.

7. PLOS authors have the option to publish the peer review history of their article (what does this mean?). If published, this will include your full peer review and any attached files.

Reviewer #1: **Yes: **Graciela Fabiana Scruzzi

Reviewer #2: No

---

## [Editor Report · Acceptance letter]

24 Aug 2023

PONE-D-23-13686R1 

Emotional overeating affected nine in ten female students during the COVID-19 University closure: A cross-sectional study in France 

Dear Dr. Val-Laillet:

I'm pleased to inform you that your manuscript has been deemed suitable for publication in PLOS ONE. Congratulations! Your manuscript is now with our production department. 

Kind regards, 

on behalf of

Dr. Roberto Ariel Abeldaño Zuñiga 

Academic Editor

PLOS ONE